# Teacher Training and Sustainable Development: Study within the Framework of the Transdisciplinary Project RRREMAKER

Pilar Manuela Soto-Solier [1], Ana María García-López [2] and María Belén Prados-Peña [3],*

1    Department of Didactics of Musical, Plastic and Corporal Expression, Faculty of Educational Sciences, University of Granada, 18071 Granada, Spain; psolier@ugr.es
2    Department of Drawing, Faculty of Fine Arts, University of Granada, 18071 Granada, Spain; agarcial@ugr.es
3    Department of Marketing and Market Research, University of Granada, 18071 Granada, Spain
*    Correspondence: bprados@ugr.es

**Abstract:** The development of the Sustainable Development Goals (SDGs) is a challenge that requires the involvement of the educational system. This study seeks to identify the perception and knowledge of future teachers in relation to sustainable development (SD), the European Green Deal (ECD), and circular economy (CE) at the University of Granada. A qualitative study was carried out, conducted face to face, using an online questionnaire in the classroom. A total of 321 students participated, from bachelor's degrees in early childhood and primary education to master's degrees in secondary education. Out of these, 176 validated questionnaires were analyzed. The results showed that future teachers understood the importance of SD education as an important social problem. They believed that education in SD, EGD, or CE could contribute to solving environmental and social problems. Nevertheless, they doubted whether this training should be included by universities in their training actions for teachers; they expressed reservations about the educational effectiveness of the curricula. However, most of them considered the need for more training in this area. Finally, they associated sustainable development mainly with the environmental dimension, followed by the social dimension, and to a lesser extent with the economic dimension.

**Keywords:** educational research; teacher training; sustainable development; quantitative research; transdisciplinarity; social responsibility

## 1. Introduction

The world situation continues to be marked by the ravages of the severe pandemic (COVID-19), which, together with catastrophes, climate change, inequalities, etc., have put different governments in check in the face of an uncertain future for humanity, highlighting the fragility of our interrelationship with nature. In this sense, UNESCO, as the organization in charge of coordinating the 2030 Agenda, together with the European Commission [1], has identified education for sustainable development (hereinafter ESD) as one of its priorities, which becomes explicit in one of the 17 Sustainable Development Goals (SDGs): the number 4 goal seeks to "ensure inclusive, equitable and quality education and provide lifelong learning opportunities" [2]. More specifically, Target 4.7 focuses on ensuring that students obtain the theoretical and practical knowledge necessary to activate sustainability-based development and improve education and livelihoods, as well as people's rights, among others [3] (p. 57). The SDGs pursue quality education focusing on three dimensions of learning, namely, cognitive, socioemotional, and behavioral dimensions [1] in formal, informal, and non-formal educational settings.

On the other hand, the Education 2030 Framework for Action [2] sets out the necessary guidelines for the acquisition of new knowledge aimed at understanding and solving problems, reflection, and the development of new ways of living [2–4]. In this context, as both researchers and teachers, we ask ourselves whether what students learn is really

meaningful to their lives. Does education provide students with strategies for their own and the planet's survival in line with UNESCO's guidelines? Is transdisciplinary education and research in arts, design, technology, and circular economy a guarantee of sustainable development?

It is a challenge for ESD to facilitate the acquisition of knowledge, awareness, and activation of actions that empower students to generate transformations in themselves and in the society around them. As the reviewed literature [5] (p. 29) indicates, reorienting education to address sustainability is a profound process that involves changes in programs, practices, and policies, as well as awareness, knowledge, skills, values, and acceptance of the sustainability paradigm.

This approach also implies the promotion of a culture of awareness and social responsibility (hereinafter SR), and the generation of social links to create fairer, more diverse, and inclusive communities [6]. This requires a transversal, multidisciplinary, and transdisciplinary training based on the active and collaborative strategies included in the 2030 Agenda. Specifically, "The New Circular Economy Action Plan" [7] and the European Green Deal [8] aim to achieve a clean and competitive Europe and a world that is more efficient in the use of resources and, in turn, more competitive. To respond to this challenge, at the national level, the "Spanish Circular Economy Strategy, Spain Circular 2030" and the "Circular Economy Action Plan 2021/2023" [1,7,8] have been designed.

In this line, the EGD [8] adds to the approach that education and training are essential to raise awareness and develop capacities to activate the green ecoeconomy, directly related to the circular economy due to its immediate consequence with the environmental problems exposed in the "EGD" and in the "Circular Economy Action Plan". With the aim of achieving a cleaner and more competitive continent by 2050 (through the creation of markets based on clean technologies and products, sustainable transport, ecological lifestyles, etc.), the EGD Plan includes the pillars of the economy of the immediate future, strategies on biodiversity, CE, zero pollution, renewable energies, etc. From a systematic approach, the CE tries to explore symbiotic ways to design circular urban systems and optimize the materials and energy metabolism of cities to minimize the environmental footprint [9]. To achieve this goal, it is necessary to design strategies and methodologies aimed at learning innovation, action, and quality education [10], which develop those personal, curricular and professional competencies, as well as "values in action" programs that contribute to building a sense of environmental and economic social responsibility [9,11].

Disruptive proposals that link research with methodologies oriented toward the paradigm shift necessary to mitigate the environmental footprint through actions that seek social, educational, and economic transformation are needed. These proposals are extensively developed in transdisciplinary research projects such as RRREMARKER (Reuse Reduce Recycle Platform based on AI (Artificial Intelligence) for an automated and scalable Maker culture in the circular economy). The extensive objective of this RISE program, called RRREMAKER, is to develop an artificial intelligence (AI)-based maker platform for the design and production of handcrafted, rapid prototyped, and reconditioned products, based on the availability of used goods and recyclable waste collected; obtaining inputs from, and connecting together, digital manufacturers and traditional crafts, designers/creative companies, and green companies; establishing a new hybrid managing model based on the communities of knowledge, ecodesign, and democratization invention; and integrating orange, sharing, and circular economies (https://www.rrremaker.com/ (accessed on 2 July 2023), which frames this study, together with the E-ARTyTECH (art-education-technology), artistic projects, STEAM education, and creative thinking for sustainable development and social and personal transformation) innovation project linked to the first one. The overall objective of this program is the training of future teachers through education and research based on arts, crafts, and technology. Learning is based on sustainable STEAM art projects, developed in formal and non-formal contexts. Research in education is based on a critical, creative, and inclusive pedagogy, promoting innovative actions and methodologies

to improve the quality of education in line with the Sustainable Development Goals (SDGs) and social and personal transformation.

### 1.1. Knowledge and Sustainable Development, Contributions from Interdisciplinary Educational Research

In a complex, diverse, digitized, and globalized society subsisting in an increasingly vulnerable and degraded habitat, educational institutions and curricula are challenged to prompt students to learn not only basic skills but also transferable skills, such as critical thinking [12], problem solving [13], and conflict resolution, to help them become "responsible global citizens" [2], i.e., "citizens of sustainability" [14].

ESD proposes a holistic student-centered education, based on a sensitive, active pedagogy aimed at social transformation and the resolution of real problems [13]. Along these lines, the concept of sustainable development (SD) in general refers to a balance between the environmental, economic, and social dimensions of sustainability, although the meaning of the social component is yet to be defined [15]. In this sense, it calls for an interdisciplinary, multidisciplinary, and transdisciplinary education that allows the generation of synergies and links between different learning and disciplines for life, enabling the development of competencies for the development of a sustainable society [16].

Different authors [17] observed in their research that active teachers have general knowledge about the three components of ESD, but they need to understand their interrelation. Another study [18] indicates that teachers are aware of the relevance of the three dimensions (social, environmental, and economic) of ESD to varying degrees but lack a holistic understanding of the SDGs' concepts [19]. On the other hand, certain studies show that teachers' knowledge is more common in the ecological perspective [20], contrasting with insufficient knowledge in sustainability [21], which hinders the implementation of ESD in the classroom.

Recent studies [22] have found that one of the most present competencies in the curriculum is to improve the application of behaviors and ethical principles at the professional and personal level in relation to the value of sustainability, while the least present is the sustainable use of resources in the environmental and social environment. The indications demand educational innovation oriented to sustainability at the curricular level [23], going beyond the academic sphere to promote change in real life.

In this line, we find research on knowledge acquisition in ESD, which shows its effectiveness in improving knowledge and environmental responsibility (hereinafter AR). Examples of this are interdisciplinary studies of engineering and design applied to sustainable design [24] or those linking co-education and co-creation processes [25], highlighting research focused on the curriculum and sustainability at the university level [26].

Likewise, as pointed out by the 2030 Agenda and the Digital Education Action Plan [27], another priority is to integrate the use of technology in the teaching and learning process, developing knowledge (disciplinary, interdisciplinary, and transdisciplinary) linked to digital literacy [28,29]. Currently, technological learning tools are changing the way we live, work, and interact in society. Novel technologies, methodologies, and approaches (STEM and STEAM) are being applied that can address complex sustainability issues. These methodologies promote ESD with a practical and creative character [30], incorporating new educational models for social transformation [31].

### 1.2. Education for Sustainable Development, Culture of Responsibility and Social Transformation

In this context, it is necessary to activate protocols of awareness and social commitment to learn how to generate social bonds and effectively manage coexistence in order to achieve a more inclusive society [6]. In this approach, it is important to investigate the knowledge of future teachers and also the perception and attitude toward education for the development of sustainability [20,32] to study new ways of sensitizing and raising awareness in society. Research should be directed to all people creating new resources and strategies to address this crisis at social, environmental, and economic levels, using innovative technological and

digital pedagogies [33] aimed at generating value on the environment around them and allowing the recognition of peers and society, understanding that we are all responsible for all [11,34].

In this line, it is necessary to investigate the knowledge of future teachers and the development of their competencies as these will be reflected in their future teaching practice and, therefore, in the way they teach their students to look for solutions to real problems [13]. According to the European Parliament (2020, 2022), social and environmental responsibility should favor the development of a more sustainable and resilient society and world, for which Circular Economy Strategies are designed, with a cross-cutting and interdisciplinary nature. Strategies that call for the training of today's consumers and producers are indicated by different investigations [9,10]. Other authors propose an approach in line with project-based learning supported by technology [35,36]. A transformative learning model of research and action that allows students to propose and evaluate projects that can be applied in real life, as well as learning in SD, addresses issues and problems that students observe in their environment, developing their personal empowerment [37].

In this scenario, this research has been conceived with the aim of inquiring into the knowledge, perception, and views of future teachers toward SD, CE, and EGD, key actors in their development and implementation, as disseminators of education for transformation. We raise these issues accordingly:

1. What is the degree of knowledge of prospective teachers regarding the SD, CE, and EGD?
2. How do prospective teachers view education for sustainable development?
3. What are the levels of understanding/knowledge of the SD of future teachers?

## 2. Materials and Methods

A total of 371 students participated in the study during academic year 2021–2022, including students of degrees in early childhood education (subjects: visual arts in childhood and didactics of visual arts) and degrees in primary education (course 1, subject: teaching and learning of the visual arts) and students of master's degrees in teaching compulsory secondary education and baccalaureate, vocational training, and language teaching at the University of Granada (subject: learning and teaching of drawing, image, and plastic arts). The final sample obtained was 176 prospective teachers, of whom 68.2% were women, 11.4% were men, and 20.5% did not indicate their gender. By age, 84.7% were under 25 years old, 14.8% were between 25 and 40 years old, and 0.6% were between 41 and 55 years old (see Table 1). With respect to their specialization, the majority of the participants were students in the infant grade (84.1%), followed by students in the primary grade with 12.5% and secondary students with 3.4% (see Table 2).

**Table 1.** Gender and age of participants.

|  | **No.** | **Percentage (%)** |
| --- | --- | --- |
| **Gender** | | |
| Women | 120 | 68.2 |
| Men | 20 | 11.4 |
| N/A | 36 | 20.5 |
| Age | | |
| Under 25 years old | 149 | 84.7 |
| Between 25 and 40 years old | 26 | 14.8 |
| Between 41 and 55 years old | 1 | 0.6 |

**Table 2.** Participant's specialization.

| Sample | Frequency | Percentage (%) |
|---|---|---|
| Student of the infant education degree | 148 | 84.1 |
| Primary grade student | 22 | 12.5 |
| Secondary grade student | 6 | 3.4 |
| Total | 191 | 100.0 |

*2.1. Instrument*

A questionnaire was designed on the basis of measures used by other authors to proceed with the data collection. A five-point Likert-type scale was used to measure the degree of knowledge. "Rate the degree of knowledge, from 1 (very low) to 5 (very high)": (1) knowledge of the circular economy, (2) knowledge of the European Green Deal, and (3) knowledge of the Sustainable Development Goals (SDGs) (See Table A1, Appendix A). To find out the degree to which education contributes to the development of the circular economy and sustainable development, a seven-point Likert scale was used. "With regard to the circular economy, rate the following statements on a scale of 1 (strongly disagree) to 7 (strongly agree)": (1) I need more knowledge/training on circular economy, (2) including circular economics education in my curricula could improve my ability to teach my students, (3) university departments preparing teachers at any level should include training in circular economy education, (4) teachers can contribute to solving environmental problems through their teaching, and (5) education allows training in circular economy knowledge (See Table A2, Appendix A). The questionnaire contained eight statements reflecting the different dimensions of sustainable development (economic, environmental or ecological, and social) (See Table A3, Appendix A). Finally, regarding the conceptual understanding of sustainable development by teachers and students, a seven-point Likert scale (1, strongly disagree to 7, strongly agree) adapted from the work [18] of Borg et al. (2014) was used. According to these authors, the understanding of the concept can be broken down into environmental, social, and economic dimensions (See Appendix A).

*2.2. Data Collection and Analysis Procedure*

The data were collected online, after the corresponding mailing to the respondents, which was carried out in person in the classroom after receiving the corresponding indications from the teacher. The data obtained were analyzed using IBM SPSS Statistics software version 28.0.1.0.

Table 2 shows that the majority of the participants were students in the infant education degree.

## 3. Results

The analysis of the level of knowledge has been differentiated according to three key aspects: the degree of knowledge about the "circular economy (CE)", about the "European Green Deal (EGP)", and about the "Sustainable Development Goals (SDGs)" (see Table 3).

**Table 3.** Mean level of knowledge in CE, EGD, and SDGs.

| | No. | Mean (Out of 5) | Standard Deviation |
|---|---|---|---|
| Circular economy (CE) | 176 | 1.89 | 0.913 |
| European Green Deal (EGP) | 176 | 1.76 | 0.944 |
| Sustainable Development Goals (SDGs) | 176 | 2.37 | 1.114 |

Figure 1 shows in general a very low degree of knowledge of the concepts analyzed, with mean values below 2 with respect to the CE and the EGP and slightly above 2 with respect to the SDGs.

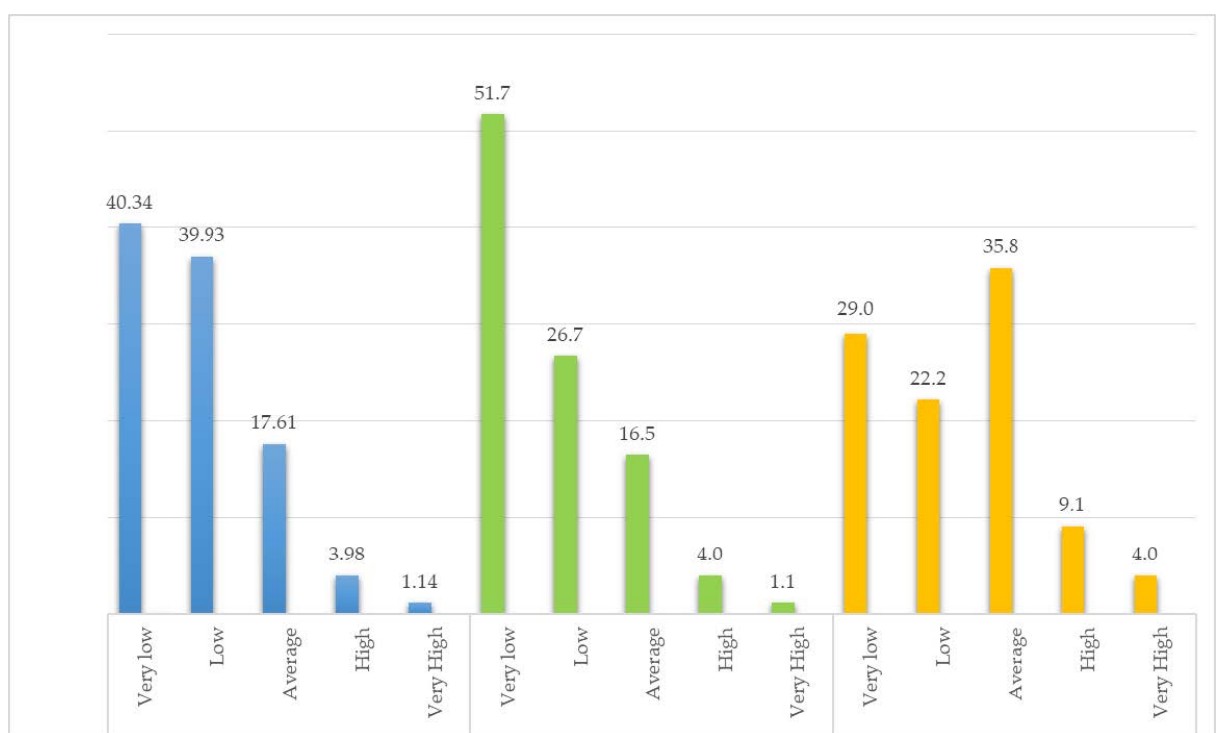

**Figure 1.** Level of knowledge of CE, EGP, and SDGs.

A more detailed analysis shows that in general, the degree of knowledge of the three issues addressed was very low and low, with the EGD being the most unknown, with 51.7% of those surveyed stating that their knowledge was very low, followed by CE with 40.3%. The SDGs were the most known, although 29.0% of the respondents expressed a very low and 4.0% a very high level of knowledge. With respect to the EGD and the CE, only 1.1% of those surveyed stated that they had a high level of knowledge (see Figure 1).

On the other hand, the degree of knowledge of each of the aforementioned topics (CE, EVP, and SDGs) was analyzed according to gender. The results obtained by means of an ANOVA showed that women showed a lower degree of knowledge in each of them than men, although the differences were not statistically significant. Thus, in the case of CE, it was observed that women showed a lower degree of knowledge than men, with mean values of 1.81 and 2.35, respectively. The same was true for the level of knowledge of the EVP (1.69 for women and 2.20 for men) and the SDGs (2.28 for women and 2.65 for men).

Next, we analyzed how prospective teachers viewed the education they received with respect to CE and SD and their contributions to the development of CE and SD. In general, 33.5% of the respondents considered that teachers could contribute to solving environmental problems through their teaching, and 37.2% indicated that they needed more training on CE (see Figure 2). However, there was no clear position as to whether the inclusion of the circular economy (CE) in the curricula could improve the capacity to teach it, as in this case, 22.0% totally agree. A slightly higher percentage, 23.6%, considered that university departments that prepared teachers should include teaching around CE (see Figure 2).

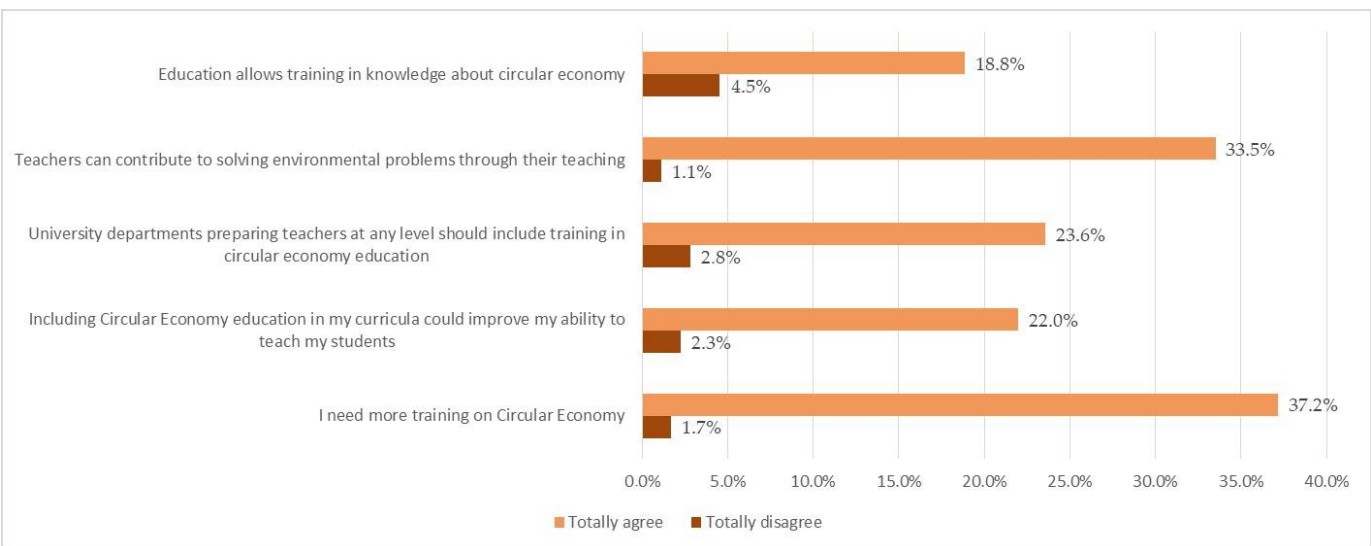

**Figure 2.** The contribution of education to the development of CE and SD.

Based on the mean values expressed by future teachers regarding the contribution of education to the development of CE, it can be observed that future teachers considered that education allowed training in CE knowledge, with a mean value of 5.6 out of 7. Next, they valued the fact that teachers could contribute to solving environmental problems through their teaching, measured with a value of d (see Table 4).

**Table 4.** Mean values for EC and education.

|  | Mean | Standard Deviation |
|---|---|---|
| I need more training on circular economy. | 5.6 | 1.567 |
| Including circular economy education in my curricula could improve my ability to teach my students. | 5.1 | 1.569 |
| University departments preparing teachers at any level should include training in circular economy education. | 5.1 | 1.546 |
| Teachers can contribute to solving environmental problems through their teaching. | 5.5 | 1.454 |
| Education allows training in knowledge about circular economy. | 5.55 | 1.567 |

Analyzing these data at a higher level of detail and considering ratings 5 and 6 as "quite agree" and 7 as "strongly agree", it was observed that 76.1% of future teachers believed that they could contribute to solving environmental problems through teaching, and 74.4% stated that they needed more training on circular economy (see Figure 3).

For the analysis of the conceptual understanding of sustainable development, the questionnaire contained eight statements reflecting the dimensions of sustainable development (economic, environmental or ecological, and social), adapted from the work of [18]. The statements had to be answered on a Likert scale (1, strongly disagree to 7, strongly agree).

Table 5 shows for each of the dimensions of sustainable development the level of understanding of each of its items.

As Table 5 shows, the economic dimension was the one least understood by future teachers. All statements related to this dimension showed the highest percentages of total disagreement. However, it was the environmental dimension that presented the highest levels of understanding in all items. In addition, as the table shows, future teachers mainly associated sustainable development with the environmental dimension in relation to "recycling of waste products", followed by the social dimension of "helping people avoid hunger and disease", with mean ratings of 5.78 and 5.71, respectively.

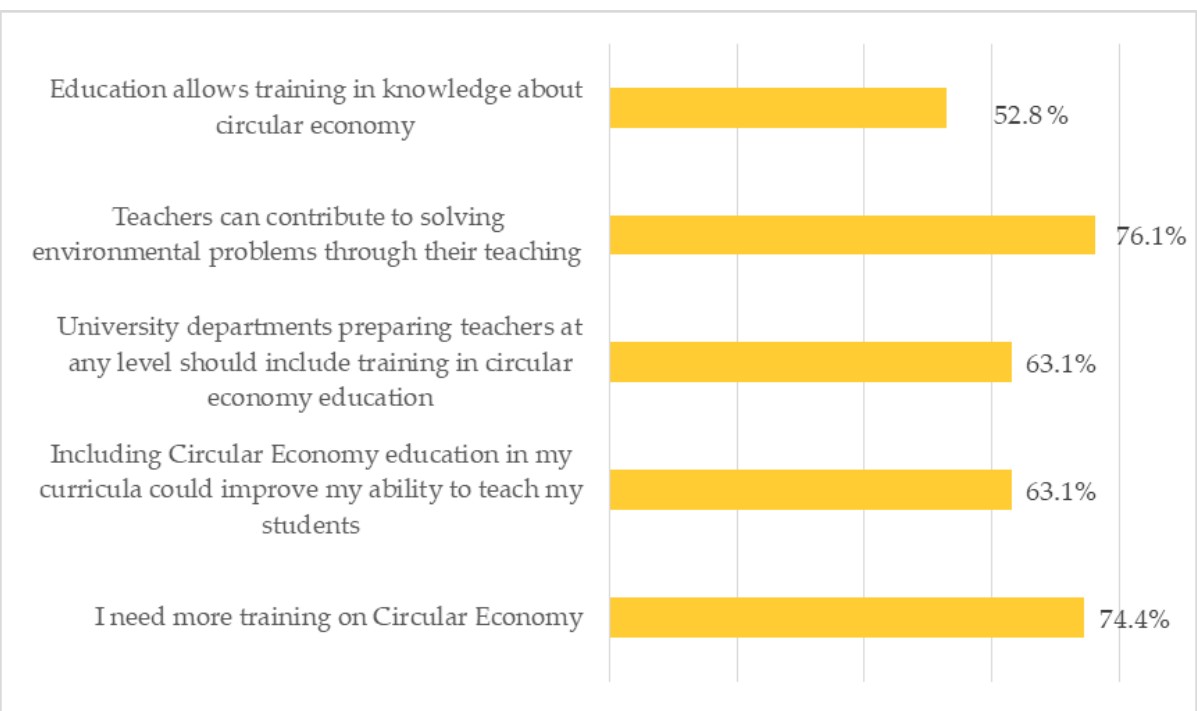

**Figure 3.** The contribution of education to the development of CD and SD. Ratings at mean and above mean (5 to 7).

**Table 5.** Conceptual understanding of sustainable development by future teachers.

| | Mean | Standard Deviation | Totally Disagree | Totally Agree | Partially Agree (Rating of 5 and 6) |
|---|---|---|---|---|---|
| **Sustainable Development Implies** | | | | | |
| **Environmental (or Ecological) Dimension** | | | | | |
| Developing new technologies to reduce the impact of harmful by-products of production | 5.31 | 1.393 | 0.60% | 26.70% | 44.30% |
| Maintaining biodiversity in the local environment | 5.44 | 1.355 | 0.00% | 29.00% | 44.90% |
| Recycling of waste products | 5.78 | 1.238 | 0.00% | 37.50% | 44.30% |
| **Social dimension** | | | | | |
| Helping people avoid hunger and disease | 5.71 | 1.398 | 0.60% | 40.30% | 36.90% |
| Social progress that recognizes the needs of all | 5.57 | 1.421 | 0% | 35.20% | 40.40% |
| **Economic dimension** | | | | | |
| Exploiting natural resources for human benefit while maintaining critical natural capital | 4.49 | 1.817 | 9.90% | 17.60% | 30.10% |
| Maintain high and stable levels of economic growth | 4.97 | 1.491 | 1.7% | 18.20% | 46.10% |
| Putting the needs of nature before those of humanity | 4.68 | 1.626 | 6.80% | 14.20% | 38.00% |

Considering the ecological dimension, "recycling of waste products" showed the highest levels of agreement with 35.7% of respondents expressing total agreement, followed by "maintaining biodiversity in the local environment" and "developing new technologies to reduce the impact of harmful by-products of production" with 29% and 26.7%, respectively (see Table 5).

Regarding the social dimension, 40.3% of respondents strongly agreed that "helping people avoid hunger and disease" was associated with the concept of sustainable development (SD), while 37.2% strongly agreed that "social progress that recognizes the needs of all" was associated with SD (see Table 5).

Finally, the economic dimension was the least recognized of the three as a concept associated with sustainable development, in its three specific manifestations, with the item related to "putting the needs of nature before those of humanity" being the item with the

lowest percentage of respondents associating it with the economic dimension of sustainable development, with only 14.2% of respondents expressing "totally agree" (see Table 5).

## 4. Discussion

The main objective of this work was to investigate the knowledge, perception, and assessment of future teachers for sustainable development at the University of Granada.

The first question raised was, *What is the degree of knowledge of future teachers with respect to the SDGs, CE, EGD?* The results showed that students that (most likely) would become future teachers had a low degree of knowledge of the analyzed concepts, with low values for CE and EGD and slightly higher values than those for *SDGs*. Coincident with the results of the research that was carried out [17], the respondents had general knowledge of the SDGs, but they lacked an understanding of their interrelationship or more specialized information. Likewise, the results suggested that future teachers were aware of the relevance of the three dimensions (social, political, and economic) of ESD to varying degrees [18], but generally, a holistic understanding of the concepts was lacking. For SDGs, as indicated by some authors [19], on the other hand, the evaluations obtained contrasted with those of other studies, which showed that teachers' knowledge tended to be more common in the ecological perspective [20] and scarcer in sustainability [21], which may hinder training in SDGs in the classroom attending to the learning of more specific concepts such as EC and EGD.

Considering the results obtained for the degrees of knowledge in CE, EGD, and SDGs, based on gender, they showed that women showed a lower degree of knowledge in each of them than men, although the differences were not statistically significant, since we must emphasize that 68.2% of the sample corresponded to women. Even if not significant (due to the slight difference in the assessment between genders), we must take into consideration that the development of knowledge and gender equality were the main SDGs challenges that we will continue investigating. Some studies discovered that knowledge about sustainability, specifically environmental knowledge, may be affected by gender, revealing that men might have more knowledge about environmental issues than women [38]; however, women show concern and more effective environmental behavior than men [39].

The second question raised was, *How is the perception of future teachers regarding education regarding CE and SD and their contribution to the development of CE and SD?*

The data showed that the majority of respondents "strongly agreed" that teachers could contribute to solving environmental problems through CE teaching. Thus, it stands out that 76.1% of future teachers believed that they could contribute to solving environmental problems through teaching. Another significant fact was that a large number of respondents expressed the need for more training and/or training on CE in educational institutions. These results showed the need to transform education and generate effective lines of research and innovation oriented toward sustainability at curricular levels [23] and at a personal and social level [13,37].

On the other hand, the results showed the need for the involvement of institutions and educational plans in the challenge of raising awareness, training, and transformation from education, to ensure that students acquire not only basic skills and abilities but also transferable ones such as reflective and critical thinking [12], solving problems [13], or action and conflict resolution, helping them to be responsible citizens in a global world (UNESCO, 22017), "citizens of sustainability" [14].

In addition, the obtained data regarding the contribution of education in the development of CE showed that the majority of future teachers considered that education allowed training in knowledge about CE. This allowed us to read along the lines of the educational challenge in SD, confirming the urgency of a transversal and multidisciplinary training based on active and collaborative strategies and significant approaches to mitigate environmental footprints [9].

The third question was, *What are the levels of understanding/knowledge of SD of future teachers?* The analysis was carried out according to each one of the dimensions of SD,

economic, environmental/ecological, and social [18]. Regarding the achieved results, the "economic dimension" was poorly understood by future teachers since the statements in relation to this dimension showed the highest percentages in "total disagreement", contrary to the environmental (or ecological) dimension, which was the one with the highest levels of understanding in all its items. Regarding the economic dimension, we highlight the evaluations of the items "exploit natural resources for human benefit while maintaining critical natural capital" and "put the needs of nature before those of humanity", in which a very high percentage of respondents "partially agreed". This indicated that possibly, the teachers did not conceptually relate the information of these items to sustainable development. The economic dimension of SD, in accordance with studies by different authors [18], was the least recognized by the participating future teachers; it was also understood as the dimension associated with the greatest uncertainty since it exhibited the greatest variation in the conceptual understanding of the teachers.

In addition, and considering the environmental (or ecological) dimension, the considerations of future teachers based on the items "sustainable development implies developing new technologies to reduce the impact of harmful by-products of production", "maintenance of biodiversity in the local environment", and "recycling of waste products", the results showed that a very high percentage of future teachers were "partially in agreement". This revealed that they mainly associated sustainable development to this dimension in relation to the "recycling of waste products", followed by the social dimension of "helping people to avoid hunger and disease", with a high rating. This coincided with the studies that urged the development of sensitive, active, and collaborative pedagogy aimed at the transformation of society and solution of real-life problems [13] since the results showed that the meaning of the social component was still not well define [15]. On the other hand, the results showed the need to involve students in social and civic participation and creation using technologies, becoming increasingly effective tools.

Bearing in mind the ecological dimension, the item that referred to "sustainable development implies recycling of waste products" presented the highest levels of respondents who stated they "totally agreed", followed by "maintenance of biodiversity in the local environment" and "developing new technologies to reduce the impact of harmful by-products of production". Assessments that coincided with certain investigations were carried out [22], revealing that one of the most present elements in the curriculum was to improve the application of behaviors and ethical principles in the professional and personal field related to the value of sustainability, while the least present was the sustainable use of resources for disaster prevention in the environmental and social environment.

Finally, considering the social dimension, we highlight that the considerations of a high percentage of those surveyed showed that they fully agreed that "sustainable development implies helping people to avoid hunger and disease" and that "social progress recognizes the needs of all". Coinciding with the results of other investigations [18], the conceptual understanding of teachers is an issue that requires further training, urging the development and acquisition of those necessary personal, curricular, and professional skills and introducing value programs, "values in action", that contribute to building a sense of citizenship and social responsibility at a relational and environmental level [9,11]. A deeper investigation that opens new lines of study with an interdisciplinary nature (education, technology, art design, engineering, economics, etc.) is required. The opinions expressed by future teachers have reaffirmed the important need for more training that leads to more training on SD, being aware that it is a deep and reflective process that requires a great commitment on the part of the institutions.

The questionnaire first evaluated the degree of general knowledge of the student in relation to CE, EGD, and SD following the recommendations contained in the UN SDGs document (https://unesdoc.unesco.org/ark:/48223/pf0000252423 (accessed on 29 June 2023). Subsequently, for the rest of the questionnaire, key aspects were presented in relation to the need to deepen their training around these concepts. Finally, in relation to the environmental, economic, and social dimensions of sustainable development, the main

items representing these dimensions were presented to assess their degree of knowledge. Obviously, the answers given by the students reflected the "feeling of knowledge" of such students and not their real knowledge.

In this environment, in order to improve the knowledge and skills of future teachers and for them to develop transversal sustainability skills, education must go beyond experience in the educational environment, proposing collaborative work between different communities, educational levels, and families. All this can be accomplished through significant interdisciplinary and transdisciplinary methodologies (STEAM methodologies, project-based learning, problem-based learning, etc., that link art, design, ecodesign, crafts, sustainability, economics, and engineering, among others) in which it is necessary to continue research to achieve education for sustainable development linked to technological and digital literacy but also to improve creative, critical, entrepreneurial, and sustainable thinking that results in better socialization and/or social transformation, tools necessary for individual and group development throughout life.

In this sense, different lines of research are opened for the development of educational strategies, instruments, and methodologies that link analog, digital–virtual, and hybrid interdisciplinary learning processes to achieve the quality and sustainability education of the 21st century that the 2030 Agenda proposes. It is also necessary to investigate strategies and instruments for an effective evaluation of new forms of knowledge acquisition, new digital resources, educational processes, and methodologies based on technology.

## 5. Limitation

Although enlightening, the results obtained in this research have limitations; one of them shows that the sample is mostly composed of future women teachers of the degree in early childhood education, which could have led to the assessments made since 97% of the students were mostly women.

On the other hand, from a future perspective, the research should be carried out on a larger scale and with more factors to examine the knowledge of EC, EGD, and SD of future teachers. The use of methodological triangulation and the application of several data collection methods would also be significant.

## 6. Conclusions

In the context of sustainable development (SD), focusing on education for sustainable development (ESD), the results of this study indicate a general low knowledge of future teachers, especially in CE and EGD, being aware of the relevance of the three social, political, and economic dimensions of ESD to varying degrees, but a holistic understanding of the concepts of the Sustainable Development Goals (SDGs) is generally lacking.

In addition, the results indicate that future teachers understand that they can contribute to solving environmental problems through the acquisition of knowledge and teaching CE and EGD; however, they are aware of the need for more training and training. The study shows that of the SD dimensions, the economic one is the least understood by future teachers, and the environmental one is the best known. Students are also aware that the environment is an important problem, but they are unaware that it is related to social and economic problems, coinciding with research carried out [19] whose results showed that future teachers also perceived environmental factors as more relevant (87%), then economic (69%) and social (49%). The ecological dimension, as in the present study, was also predominant in relevant studies carried out with university teachers [20], as well as in inquiries about the conceptual understanding of active teachers [18]. This overvaluation of the environmental dimension could be due to the connection that many make between ESD and environmental education, making necessary an effective educational approach that implements the relational dimensions [9,11]. Regarding the differences in terms of gender, the results obtained are not conclusive, although they are in line with other investigations, in which gender differences are found.

Furthermore, the results show that future teachers do not have a clear position regarding the inclusion of CE and EGD in the study plans, which we consider an indicator of the need for greater involvement on the part of institutions and educational plans in the challenge of raising awareness, training, and transformation from education to achieve the SDGs proposed by the 2030 Agenda.

The results of this study show us the need to continue researching and deepening the perception and knowledge of future teachers in the degrees of early childhood education, primary and master's degrees in compulsory secondary teaching, and baccalaureate and vocational training in other universities of Andalusia and Spain. These results allow us to open new lines of interdisciplinary research and education, with the challenge of designing strategies and methodologies aimed at learning innovation, action, and quality education [10] and the development of those personal, curricular, and professional skills, as well as "values in action" programs that contribute to building a sense of environmental and economic social responsibility [9,11].

The information obtained also allows us to propose new disruptive proposals that link research with methodologies aimed at the necessary paradigm shift to mitigate the environmental footprint through actions that seek social, educational, and economic transformation. Based on these lines of inquiry, interdisciplinary research projects such as RRREMAKER (https://www.rrremaker.com/ (accessed on 2 July 2023) are developed, which frames this study, together with the E-ARTyTECH innovation project linked to the first. These projects are focused on studying and proposing learning environments, knowledge transfer, and meeting places between researchers and professionals of recognized prestige in education for sustainability, education, and research based on the arts, contemporary art, design, crafts, economics, engineering, and technologies associated with the craft sector, as well as undergraduate and postgraduate students (formal education) and other people such as the artisan collective (non-formal education).

**Author Contributions:** Conceptualization, P.M.S.-S., A.M.G.-L. and M.B.P.-P; methodology, P.M.S.-S. and M.B.P.-P.; validation, P.M.S.-S., A.M.G.-L. and M.B.P.-P.; formal analysis, P.M.S.-S., A.M.G.-L. and M.B.P.-P.; investigation, all authors; resources, all authors; data curation, M.B.P.-P; writing—original draft preparation, P.M.S.-S., A.M.G.-L. and M.B.P.-P. writing—review and editing, all authors; visualization and supervision, P.M.S.-S., A.M.G.-L. and M.B.P.-P.; project administration, P.M.S.-S., A.M.G.-L. and M.B.P.-P.; funding acquisition, P.M.S.-S. and A.M.G.-L. All authors have read and agreed to the published version of the manuscript.

**Funding:** This research is carried out and financed within the framework of the European Project RRREMAKER: Re-use, Reduce, Recycle. Artificial Intelligence (AI) based platform for automated and stable Maker culture in the circular economy. It is a project funded by the European Union within the MarieSktoowska-Curie Actions Program. Research and Innovation Staff Ex-change (Rise). Call: H2020-MSCA-RISE-2020. GA n° 101008060 (https://www.rrremaker.com/) (accessed on 2 July 2023), together with the E-ARTyTECH Innovation Project artistic projects, STEAM education and creative thinking for the sustainable development and social and personal transformation. (Code 22-137), under the "Call for Teaching Innovation Projects of the FIDO University of Granada Plan (Spain) 2022–2024", linked to the first.

**Informed Consent Statement:** The Research Ethics Commission of the University of Granada (Spain), having seen the mandatory report issued by the President of the Committee on Human Research, after the collegiate assessment of the Committee in plenary session, in which it is made certify that the proposed research respects the principles established in international and national legislation in the field of biomedicine, biotechnology and bioethics, as well as the rights derived from the protection of personal data, Issues a Favorable Report in relation to the investigation being registered with the number: 3542/CEIH/2023.

**Data Availability Statement:** The data used in this project has been compiled with the approval of the Research Ethics Commission of the University of Granada, registered under number: 3542/CEIH/2023. The data may be made available to interested readers by formal correspondence to the Human Ethics Research Committee of the University of Granada (Spain), registered under number: 3542/CEIH/2023.

**Conflicts of Interest:** The authors declare no conflict of interest.

**Appendix A**

*Questionnaire*

Good afternoon days,

We thank you for your collaboration in this study that we are carrying out on the Circular Economy/Sustainability. This survey tries to find out the degree of knowledge, attitude, training and assessment of the students of the Degree in Education and the active teachers themselves (both at the university and non-university educational levels) in Circular Economy, including issues related to the Sustainable Development Goals and the European Green Deal. We ask you to read carefully and to answer the questions asked honestly. Remember that there are no right or wrong answers, it is about your opinion on different issues. Remember also that your answers are anonymous and that total confidentiality of the data is guaranteed.

This study is part of the European project RRREMARKER: Reuse Reduce Recycle Platform based on AI (Artificial Intelligence) for an automated and scalable Maker culture in the circular economy.

This project is funded by the EU within the Marie Skłodowska-Curie Actions programme. Research and Innovation Staff Exchange (RISE). Call: H2020-MSCA-RISE-2020, led by the UGR.

Thank you very much for your help.

**Table A1.** Regarding the circular economy, assess the degree of knowledge. From 1 (not at all) to 7 (very high).

| Knowledge of the Circular Economy | 1 | 2 | 3 | 4 | 5 | 6 | 7 |
|---|---|---|---|---|---|---|---|
| Knowledge European Green Deal | 1 | 2 | 3 | 4 | 5 | 6 | 7 |
| Knowledge of the Sustainable Development Goals (SDGs) | 1 | 2 | 3 | 4 | 5 | 6 | 7 |

**Table A2.** Regarding the circular economy, assess the following statement. 1 strongly disagree to 7 strongly agree.

| I Need More Knowledge/Training on Circular Economy | 1 | 2 | 3 | 4 | 5 | 6 | 7 |
|---|---|---|---|---|---|---|---|
| Including Circular Economy education in my curricula could improve my ability to teach my students | 1 | 2 | 3 | 4 | 5 | 6 | 7 |
| University departments preparing teachers at any level should include training in circular economy education | 1 | 2 | 3 | 4 | 5 | 6 | 7 |
| Teachers can contribute to solving environmental problems through their teaching | 1 | 2 | 3 | 4 | 5 | 6 | 7 |
| Education allows training in circular economy knowledge | 1 | 2 | 3 | 4 | 5 | 6 | 7 |

**Table A3.** Sustainable development implications.

| Developing New Technologies to Reduce the Impact of Harmful By-Products of Production | 1 | 2 | 3 | 4 | 5 | 6 | 7 |
|---|---|---|---|---|---|---|---|
| Maintaining biodiversity in the local environment | 1 | 2 | 3 | 4 | 5 | 6 | 7 |
| Recycling of waste products | 1 | 2 | 3 | 4 | 5 | 6 | 7 |
| Helping people avoid hunger and disease. | 1 | 2 | 3 | 4 | 5 | 6 | 7 |
| Social progress that recognizes the needs of all | 1 | 2 | 3 | 4 | 5 | 6 | 7 |
| Exploiting natural resources for human benefit while maintaining critical natural capital | 1 | 2 | 3 | 4 | 5 | 6 | 7 |
| Maintain high and stable levels of economic growth. | 1 | 2 | 3 | 4 | 5 | 6 | 7 |
| Putting the needs of nature before those of humanity | 1 | 2 | 3 | 4 | 5 | 6 | 7 |
| Developing new technologies to reduce the impact of harmful by-products of production. | 1 | 2 | 3 | 4 | 5 | 6 | 7 |

Sociodemographic profile:

- Gender
- Age
- Role: infant, primary and secondary student or infant, primary and secondary teacher
- University of belonging:
- School
- Concerted or Public
- City where he teaches

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
