# Peer review of "Teacher Training and Sustainable Development: Study within the Framework of the Transdisciplinary Project RRREMAKER"

_education, doi:10.3390/educsci13080794_

Round 1

Reviewer 1 Report

Although reference 18 used such questionnaire and has been published, I suggest to mention in 4 discussion or 5 Limitations the following theoretical problem in fig 1 and Table 3: if a student does not know a field of knowledge (e.g. Circular economy), how can he/she estimate the own knowledge (the unknown subset of a hardly known set of things)? Some selected knowledge-test questions would give a better estimation on the knowledge of students. So the results refer to the 'feeling of knowledge' of students, and not to their real knowledge.

Please use English abbreviations coherently, define the used abbreviation at the first occurence.

Line 13 PVE is probably the European Green Deal (and not 'Pack'), correct it in lines 13, 60, 63 66 151 155 219 227 230 340 342 351 352 444; PV in 464,

Instead of Pact in line 7, 56, 177, 214, 223, 504 use Deal.

Line 148, 282, 350, 386 and 387: DS (must be SD, English: sustainable development), should also be used as the abbreviation of the English equivalents, 

CD and DS, ODS (Objectivos de Desarrollo sostenible?) must be used and defined in English as well. 

Line 117 Agenda 2030.

Line 203 Percentage.

Line 286 adjective 'different' should be deleted.

line 410 instead of 'resolution': solution.

Author Response

Dear reviewer,

we have attached a revised version of the manuscript, which includes a number of relevant modifications reflecting the comments, suggestions and recommendations presented by the review team. We have commented on each piece of feedback separately in this response document.

We will provide detailed answers to all the questions raised in the following sections. All updated content in the manuscript has been highlighted in BLUE.

Comments R1-1. Although reference 18 used such questionnaire and has been published, I suggest to mention in 4 discussion or 5 Limitations the following theoretical problem in fig 1 and Table 3: if a student does not know a field of knowledge (e.g. Circular economy), how can he/she estimate the own knowledge (the unknown subset of a hardly known set of things)? Some selected knowledge-test questions would give a better estimation on the knowledge of students. So the results refer to the 'feeling of knowledge' of students, and not to their real knowledge.

[REPLY R1-1]

We would like to thank the reviewer for the careful reading. We agree with the reviewer and, following the indications provided, we have included the following text in the discussion section. The text reads as follow:

The questionnaire first evaluated the degree of general knowledge of the student in relation to CE, EGD and SD following the recommendations contained in the UN SDGs document (https://unesdoc.unesco.org/ark:/48223/pf0000252423).

Subsequently, for the rest of the questionnaire, key aspects were presented in relation to the need to deepen their training around these concepts.

Finally, in relation to the environmental, economic and social dimensions of sustainable development, the main items representing these dimensions were presented to assess their degree of knowledge.

Obviously, the answers given by the students reflect the 'feeling of knowledge' of such students, and not to their real knowledge.

Comments R1-2. Please use English abbreviations coherently, define the used abbreviation at the first occurence.

Line 13 PVE is probably the European Green Deal (and not 'Pack'), correct it in lines 13, 60, 63 66 151 155 219 227 230 340 342 351 352 444; PV in 464,

Instead of Pact in line 7, 56, 177, 214, 223, 504 use Deal.

Line 148, 282, 350, 386 and 387: DS (must be SD, English: sustainable development), should also be used as the abbreviation of the English equivalents,

CD and DS, ODS (Objectivos de Desarrollo sostenible?) must be used and defined in English as well.

Line 117 Agenda 2030.

Line 203 Percentage.

Line 286 adjective 'different' should be deleted.

line 410 instead of 'resolution': solution.

[REPLY R1-1]

One again, we would like to thank the reviewer for the careful reading. We agree with the reviewer and, following the indications provided, we have corrected the abbreviations and have unified them in the text.

For the abbreviation from European Green Deal (not European Green Pact), we have used EGD.

For the abbreviation from Sustainable Development (SD) we have corrected in the text DS for SD.

The abbreviation ODS  has been changed to (SDGs -  Sustainable Development Goals).

For Agenda 2030, we have changed to 2030 Agenda.

We have changed “percentage”.

We have deleted “different” in the text.

We have changed “resolution” for solution.

Reviewer 2 Report

In the abstract, the authors use several abbreviations, which are correctly entered. However, for one of them, namely European Green Pact 7 (ECP), the authors also use another abbreviation - PVE, probably derived from the Spanish version - El Pacto Verde Europeo. To ensure unity, authors should keep the abbreviation from the English name of the concepts used, both in the abstract and in the content of the work.

For the abbreviation from Sustainable Development (SD) introduced in the abstract, the authors also use another abbreviation within the article, namely DS. This should be corrected.

The abbreviation ODS that first appears on page 5 is not explained. Then it appears several times in the work.

 In the introduction, the authors mention some research projects, named only by their acronyms (RRREMAKER and E-ARTyTECH), but they do not mention the extended name of the projects, nor do they provide details about them in the work. This remains unclear to the reader. Furthermore, the title of the article includes this acronym. The article title is too long, it should be shortened.

 Some sentences seem incomplete, as for example”Strategies that call for the training of today's consumers and producers [9, 10]” (page 3, lines144-145) or ”An approach in line with Project Based Learning supported by technology” (page 3, lines 145-146).

Page 4, line 170: The authors wrote "(See Table 2). (See Table 2)" twice.

Page 9, line 315: The authors wrote twice "(see Figure 4 and Table 5). (See Figure 4 and Table 5)".

 On pages 4 and 5 there are two paragraphs that present the same idea.

On page 4, the authors wrote:

"With respect to their specialization, the majority of the participants were students in the infant grade (84.1%); followed by students in the primary grade with 12.5% and secondary students with 3.4% (See Table 2). (See Table 2).”

On page 5, the authors wrote:

"Table 2 shows the participant profile. The majority of the participants were students in the Infant Grade (84.1%); followed by students in the Primary Grade with 12.5% and high school students with 3.4%."

Authors should keep only one of the paragraphs.

In Figure 2, the percentage of 37.2% is not visible because it overlaps the bar in the figure. The same problem appears in the case of several percentages in Figure 3.

Table 5 and Table 6 could have been unified as both present data related to the conceptual understanding of sustainable development by future teachers.

Table 5 and Figure 4 duplicate the data. In addition, some percentages in Table 5 and Figure 4 do not correspond.

The same problem is in the case of Table 5 and Figure 5, respectively in the case of Table 5 and Figure 6. The authors should keep only one way of representing the data (either figure or table) and check the percentages one more time.

A major problem of this article is related to the fact that the authors consider that they have provided answers related to the knowledge of prospective teachers regarding the Sustainable Development, Circular Economy, the European Green Pact (see the purpose of the research and research questions 1 and 3). In fact, the authors only present perceptions that future teachers have about the knowledge related to these aspects. Perceptions about knowledge and knowledge itself are not the same thing. This aspect should be corrected by the authors and presented as a limitation of their research.

Also in the limits category, the authors should have specified the use of a single research method. Methodological triangulation is not ensured.

In the discussion section, the authors should have proposed several concrete ways of training future teachers regarding Education for Sustainable Development, but this aspect is not detailed enough.

The research ethical issues are not presented by the authors.

Author Response

Dear reviewer,

we have attached a revised version of the manuscript, which includes a number of relevant modifications reflecting the comments, suggestions and recommendations presented by the review team. We have commented on each piece of feedback separately in this response document.

We will provide detailed answers to all the questions raised in the following sections. All updated content in the manuscript has been highlighted in BLUE.

Best regards

Round 2

Reviewer 2 Report

At the first mention of the abbreviation in the abstract, the authors wrote ”the European Green Deal (ECP)”. It should have been written (EGD).

Page 9: The authors should remove the following sentence from the manuscript because it is a comment for the reviewer: "By unifying the tables, we have redrafted the paragraphs that referenced them and they have been as follows".

Author Response

Response to Reviewer 2

We have attached a revised version of the manuscript, which includes a number of relevant modifications reflecting the comments, suggestions and recommendations presented by the review team. We have commented on each piece of feedback separately in this response document.

We will provide detailed answers to all the questions raised in the following sections. All updated content in the manuscript has been highlighted in BLUE.

Comments R2-1. At the first mention of the abbreviation in the abstract, the authors wrote  ”the European Green Deal (ECP)”. It should have been written (EGD).

Page 9: The authors should remove the following sentence from the manuscript because it is a comment for the reviewer: "By unifying the tables, we have redrafted the paragraphs that referenced them and they have been as follows".

[REPLY R2-1]

We would like to thank the reviewer for the careful reading. We agree with the reviewer and, following the indication provided, we have corrected the mistakes we have made.
